# Assessment of Sponge City Flood Control Capacity According to Rainfall Pattern Using a Numerical Model after Muti-Source Validation

Haichao Li [1,*] , Hiroshi Ishidaira [1] , Yanqi Wei [2], Kazuyoshi Souma [1] and Jun Magome [1]

[1] Interdisciplinary Centre for River Basin Environment, University of Yamanashi, Kofu 400-8511, Japan; ishi@yamanashi.ac.jp (H.I.); ksohma@yamanashi.ac.jp (K.S.); magome@yamanashi.ac.jp (J.M.)
[2] School of Energy and Power Engineering, Xihua University, Chengdu 610039, China; weiyanqi233@gmail.com
* Correspondence: g19dtka3@yamanashi.ac.jp; Tel.: +81-070-4352-1786

**Abstract:** Urban floods are a common urban disaster that threaten the economy and development of cities. Sponge cities can improve flood resistance ability and reduce floods by setting low-impact development measures (LID). Evaluating flood reduction benefits is the basic link in the construction of sponge cities. Therefore, it is of great significance to evaluate the benefits of sponge cities from the perspective of different rain patterns. In this study, we investigated the urban runoff of various rainfall patterns in Mianyang city using the Strom Water Management Model (SWMM). We employed 2–100-year return periods and three different temporal rainfall downscaling methods to evaluate rain patterns and simulate urban runoff in Mianyang, with and without the implementation of sponge city measures. After calibration, model performance was validated using multi-source data concerning flood peaks and inter-annual variations in flood magnitude. Notably, the effects of peak rainfall patterns on historical floods were generally greater than the effects of synthetic rainfalls generated by temporal downscaling. Compared to the rainfall patterns of historical flood events, the flood protection capacities of sponge cities can be easily overestimated when using the synthetic rainfall patterns generated by temporal downscaling. Overall, an earlier flood peak was associated with better flood sponge city protection capacity. In this context, the results obtained in this study provide useful reference information about the impact of rainfall pattern on urban flood control by LID, and can be used for sponge city design in other part of China.

**Keywords:** SWMM; low-impact development; satellite observations; temporal downscaling

## 1. Introduction

Urbanization has greatly increased in the past century. As of 2011, the global urbanization rate was 52.1%; it will reach 67.2% by 2050 [1]. The proportions of populations affected by floods are also increasing [2]. China is one of the world's most urbanized countries; this urbanization is expected to become more pronounced [3]. Heavy rains and floods compromise urban health. Often, drainage systems are old, while new urban areas have changed the original runoff pattern; moreover, the population is concentrated, rendering flooding and waterlogging problems increasingly serious. In recent years, waterlogging has become more common [4]. In 2012, heavy rain on 21 July in Beijing caused 10,660 houses to collapse; 1.602 million people were affected and the direct economic loss exceeded $ 1.84 billion [5]. In 2016, a rainstorm on 6 July in Wuhan affected 757,000 people and caused direct economic losses of $36 million [6]. In 2020, heavy rain on 22 May in Guangzhou caused the suspension of the subway and great economic losses [7]. In July 2021, Zhengzhou (Henan) was affected by a severe rainstorm that killed 51 people and caused direct economic loss of $10.4 billion [8]. Urban flooding has become chronic in Chinese cities, severely restricting development. It is important to strengthen research

concerning urban rainstorms and flooding, while developing prevention and mitigation measures [9].

Urban construction expands impervious surfaces and thus reduces the area available for water retention; stormwater runoff exceeds the drainage capacity [10,11], so that the rapid urbanization process has led to worsening urban flooding [12]. Flooding is sudden (caused by local heavy rain during strongly convective weather), socially impactful (causing major loss of life and property, as well as social unrest), and chaining (damaging the entire drainage system via flooding of key points or key surfaces). It is difficult to study urban flooding in an experimental manner. Urban storm/flood simulations provide the scientific basis for flood control. The models include the Storm Water Management Model (SWMM) [13], MIKE SHE [14], Soil and Water Assessment Tool (SWAT) [15], and the Institute of Hydrology Distributed Model (IHDM) [16]. For example, Merhawi et al. simulated urban flood inundation and recession affected by manholes [17], while Wu et al. [18] simulated urban flooding by coupling the SWMM and LISFLOOD-FP; Bai et al. [10] used the SWMM to study low-impact development (LID). Xu et al. [19] developed a new and general method for blockscale LID-BMPs planning, which incorporates site-scale LID-BMP chain layout optimization and block-scale scenario analysis into the planning procedure in SWMM to improve the computational efficiency and the solution quality. Yin et al. [20] propose a modelling framework of integrated one-dimensional (1D) and two-dimensional (2D) hydrodynamic modelling to evaluate the effectiveness of sponge city construction at community scale. Although simulation is efficient, its reliability depends on accurate data from local hydrological monitoring stations. There is only one verification method; this lacks versatility. Remote sensing technology detects targets at great distances; it efficiently yields accurate hydrological data. Recently, remote sensing has been used to monitor floods in small river basins and to plan water resource allocation; however, it has seldom been used to study urban flood management. The satellite data are verified by ground hydrological stations, which maximize accuracy and reliability [21]. Therefore, methods based on multi-source observations should be considered.

To effectively control and mitigate urban flooding, in December 2013, Chinese President Xi Jinping launched the concept of sponge cities to comprehensively address water scarcity and pollution, as well as urban flooding [22–24]. Sponge cities take LID as the starting point, connecting all parts of the city's water system [25,26]. Runoff and pollution caused by heavy rain are managed via decentralized, small-scale control mechanisms; the destructive impacts of development on hydrological conditions are mitigated. Developers have key roles in urban flood control in China because they are responsible for urban rainwater flood management; this responsibility is a key Chinese policy [27]. Thirty pilot sponge cities were approved in 2015; great progress has been made in terms of urban flood control, but the specific construction measures are not yet fully defined. Sponging must be quantified during planning and before construction.

How to quantify the benefits of sponge cities has always been a hot and difficult issue in research. Shao et al. quantified the impact of urbanization on flooding [28]. Simth et al. studied the hydrological response spectrum during storms in urban watersheds [29]. Fu et al. studied the impact of permeable paving in an LID area on stormwater runoff [30]. Zhang et al. studied the outcomes of green infrastructure [31]. Feng et al. studied the effects of LID measures on peak flood reductions according to return period [32]. Zhu et al. [33] conducts a life cycle, environmental, and economic quantification comparison of urban runoff source control facilities through construction and operation stages in two urban functional regions. Leng et al. [34] proposes an integrated assessment framework of coupled green–grey–blue systems on compliance of water quantity and quality control targets in Sponge City construction, where rainfall runoff and river system models are coupled to provide quantitative simulation evaluations of a number of indicators based on land and river quality. However, the rainfall data used in most studies is a single Chicago rain pattern, and the relevant characteristics of actual rainfall include rain intensity, peak occurrence time, number of peaks, etc., for a single rain pattern lacks reliability, the effect

of different rain patterns (peak occurrence time, rain intensity, and duration) on peak flood reduction in sponge cities needs to be further confirmed, and the different characteristics of these rain patterns can trigger different degrees of urban flooding. The study of the effects of different rain patterns on the flood control capacity of sponge cities will help to have subsequent decisions on sponge city construction and is important for improving the technology and layout of sponge city projects.

Furthermore, how to validate the urban-scale rainfall flood management models is also challenging, because the surface hydrological data are often lacking. Li et al. used the average runoff pollution level to explore water quality [35]. Zhao et al. converted simulated runoffs to water depths and compared them with the depths of submersion [36]. Although both methods directly or indirectly measured urban water quantity, inter-annual variation was not considered. The satellite imagery-based hydrological model established by Mark et al. accurately measures observed flows [37]. Multi-source validation improves the accuracy of the hydrological model; combinations of satellite observations with water balance measures should be considered.

Mianyang is located in the middle reaches of the Fujiang River, one of the main tributaries of the Yangtze River that is in the middle of the city near the confluence of the Fujiang River and the Anchang River. Although Mianyang is not a pilot sponge city in China, it has been selected as a "Science and Technology City" and is a local government supported by the Sichuan government for sponge city construction. The city has a warm and humid subtropical monsoon climate. The average annual temperature is 14.7–17.3 °C with an average annual precipitation of 826–1417 mm. The number of rainfall days is 195, the rainy season is mainly concentrated in June to September, and it is prone to short duration and high intensity rainstorm events. Urbanization has increased the likelihood of heavy rainfall occurring in central urban areas, such as the 7.23 mega-storm in Mianyang in 2010 and the 8.22 mega-storm in Mianyang in 2018, with most of the storm centers located in central urban areas of Mianyang.

In order to investigate the benefits of sponge city under different rain patterns to flood control, this study takes Mianyang city of Sichuan province as an example, simulates flood runoff under different rain patterns using SWMM, constructs a sponge city LID model according to Mianyang city sponge city planning, and rates and verifies the model by studying the sponge city flood control effect and rainfall process in different return periods and using satellite technology and water balance equation. The objectives of this study are: (1) to establish an urban flood simulation model applicable to Mianyang city; (2) to compare the benefits of sponge cities under different rainfall types. The aim is to provide a reference for urban scale rainfall and flood management models.

## 2. Materials and Methods

We collected hydrological, pipe network, and subsurface data regarding central Mianyang. We then constructed an SWMM to simulate actual runoff conditions; we calibrated and validated the model using two different methods. Rainfall data for different return periods were processed using three different temporal downscaling methods to assess the impacts of different patterns on the flood control capacities of sponge cities. The principal steps were (Figure 1):

(1) Dataset preparation. The hydrological data include precipitation, evaporation, and river flows. The pipe network data were collected from a drainage map provided by the local government, while the subsurface data were land use and topography.

(2) Model validation. The SWMM outputs were converted into runoff depths and the water balance method was used to quantify floods. Passive microwave remote sensing was employed to measure surface inundation; the data were used to define the dynamic trends of historical floods.

(3) Rainfall temporal downscaling. Three different downscaling methods were used to obtain rainfall patterns at different rainfall intensities, along with flood coefficients and numbers to evaluate their effects on the flood control capacity of sponge cities.

(4)	Sponge City Simulation. The impact of four LID combinations on the runoff control in the central city of Mianyang was simulated in conjunction with the sponge city planning of Mianyang.

(5)	Assessment of Flood Reduction Effect. Flood peak and volume are used as output variables to compare and analyze the abatement effect of sponge cities on urban flooding under the action of different return periods and different rain patterns. The flow chart is shown in Figure 1.

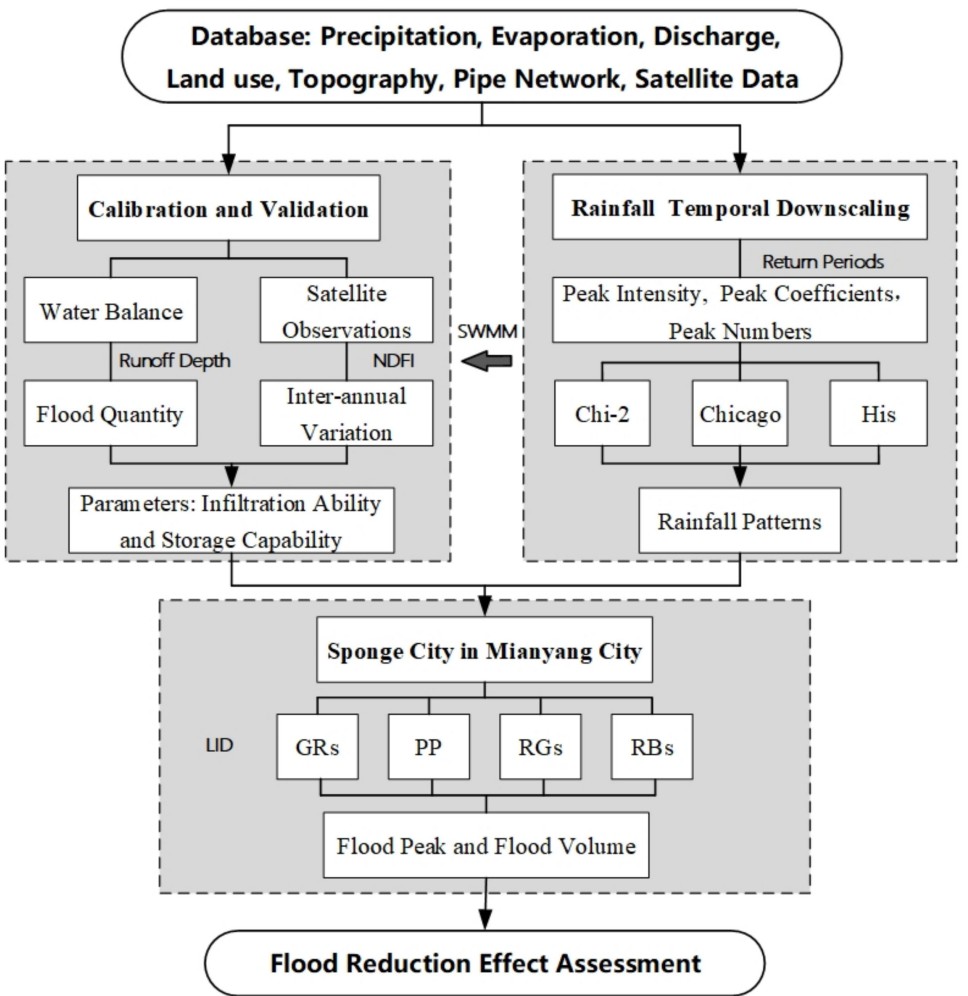

**Figure 1.** Flowchart of assessment of Sponge City Flood Control capacity based on multi-source validation and depends on different rainfall patterns.

*2.1. Study Area*

This study is about the central city of Mianyang City, Sichuan Province (Figure 2), which includes Fucheng District and Youxian District of Mianyang City, with an area of 209.2 km$^2$ and a population of about 4.8 million, while the population of the central city is about 1.8 million. The topography is high in the north and low in the south, high in the east and west, and low in the middle (altitude 450–538 m), with Fujiang River (from northeast to southwest), Anchang River (from west to east), and Furong Creek (from east to west). The average slope in the region is 6.5%, the maximum slope is 10.3%, and there is a large area of shallow hills and more obvious slope changes that are distributed in strips.

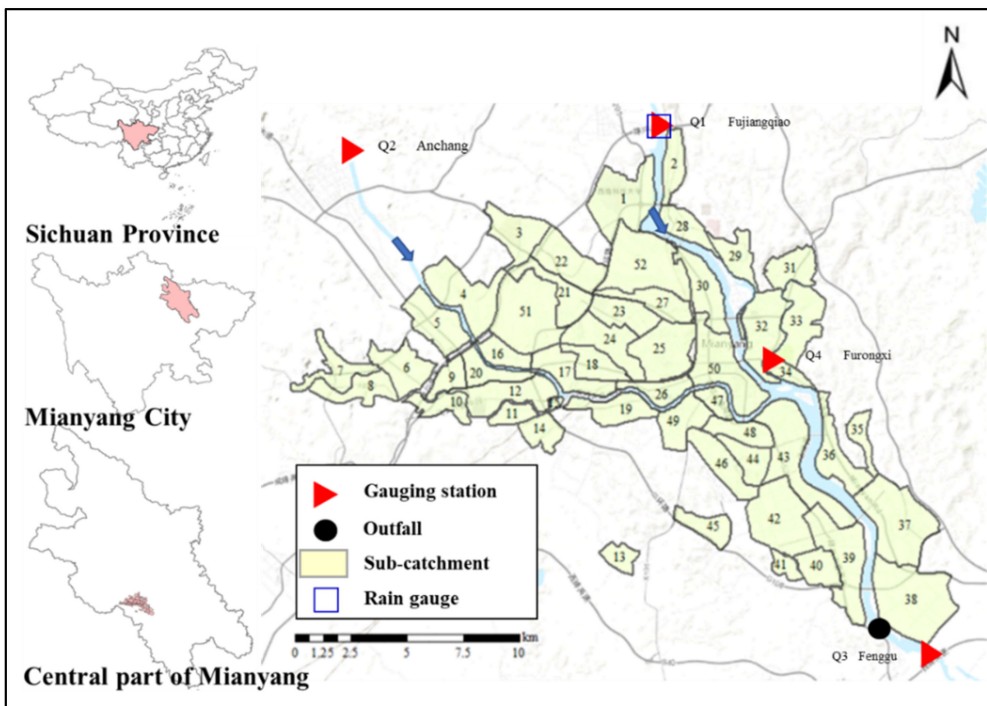

**Figure 2.** Location map of study area: central part of Mianyang City.

### 2.2. Database

The hourly rainfall observations and river flows were collected from the Sichuan hydrological business platform [Fujiangqiao Station (2015–2020)]; the daily rainfall records were collected from the Global Surface Summary of the Day [GSOD (1973–2017)]; and the monthly evaporation data were collected from the Mianyang Meteorological Bureau (2015–2020). The geographical features (areas, widths, slopes, shapes, lengths, and offset heights) of all sub-catchments were collected from the Advanced Spaceborne Thermal Emission and Reflection Radiometer (ASTER-GDEM) 30-m resolution digital topography GIS database. The land use map was based on a satellite image of Sentinel-2B (MSI Level-1C; 10-m spatial resolution; acquired 11 July 2017) that was subjected to supervised image classification. The land use types were water, reservoir, forest, building, road, cropland, and grassland. According to GlobeLand 30 [38], the percentage of urban area was 34.4% in 2010 and 65.3% in 2020. The percentage of urban area in 2017 was 54.4% according to Sentinel-2B. The result ensures the accuracy of the land use classification to a certain extent; the areal percentages of impervious regions in each sub-catchment were calculated based on the map. The pipe network was simplified from the map for downtown Mianyang (2017). The SSM/I data were provided by the National Ice and Snow Data Center in the form of cylindrical EASE-Grid projections with a resolution of 25 km, resampled to 0.25°. The details are shown in Table 1.

### 2.3. Configuration of the Urban Runoff Model

The SWMM was used to simulate the impacts of climate change and urbanization on flood control, assuming that LID practices were in effect. We used the SWMM for Mianyang City of Li et al. [39]. The study area (209.2 km$^2$) was divided into 52 sub-catchments based on topography, the pipe network, community boundaries, land use, the underlying surface, slope direction, and extent of green cover. Rainfall runoff was simulated at 15-min intervals. Certain sensitive model parameters (depression storage in impervious areas and infiltration parameters) were based on the flooding data for 2010. We confirmed that the calibrated model reasonably simulated urban runoff. In the model, each sub-catchment is

conceptualized as a rectangular surface with a uniform slope s and width W. The water balance and surface runoff are calculated as follows [40]:

$$\frac{\mathrm{d}V}{\mathrm{d}t} = A\frac{\mathrm{d}h}{\mathrm{d}t} = Ar_s - Q \tag{1}$$

$$Q = W\frac{1}{n}(h - h_p)^{5/3}s^{1/2} \tag{2}$$

where $V$, $A$, and $h$ are the storage volume, storage area, and water depth of the sub-catchment, respectively; $r_s$ is the surface runoff rate (calculated from the precipitation, evaporation, and infiltration); $Q$ is the slope outflow rate; $n$ is the Manning roughness coefficient; and $h_p$ is the depression storage depth. Each sub-catchment featured impermeable areas (road or urban land-use types) and permeable areas (forest, cropland, bare land, or water bodies); infiltration in permeable areas was calculated using the Horton equation. The parameters $n$ and $h_p$ are given for each land-use type; these parameters were then weight-averaged over the impermeable and permeable areas (depending on their proportions). Then, the ratio of $n$, and $h_p$ for impermeable and permeable areas were assigned to each sub-catchment as model parameters.

**Table 1.** Data list for SWMM and validation.

| Item | Data Source etc. | Function/Derived Features/Parameters |
|---|---|---|
| Precipitation | GSOD (Daily 1973–2017) https://www.ncei.noaa.gov/access/search/data-search/global-summary-of-the-day (accessed on 18 January 2022) Fujiangqiao Rain-gauge (Hourly 2015–2020) | Time Series, Validation |
| Evaporation | Mianyang Weather station (Monthly 2015–2020) | Monthly Evaporation |
| Discharge | 4 Hydrological stations (Hourly 2000–2020) | Validation |
| Topography | ASTER-GDEM (30 m resolution digital elv.) https://asterweb.jpl.nasa.gov/GDEM.asp (accessed on 18 January 2022) | Flow direction, Slope (gradient) |
| Land use | Sentinel-2B (10 m resolution) https://scihub.copernicus.eu/ (accessed on 18 January 2022) | Manning Coeff., Permeability, Underlying surface, Green cover |
| Pipe Network | Printed map of pipe network | Connection between each sub-catchment |
| Satellite Data | SSM/I (25 km 1991–2020) https://nsidc.org/data/NSIDC-0032/versions/2 (accessed on 18 January 2022) | Validation |

### 2.4. Multi-Source Validation

Two different methods were used to validate and calibrate the parameters set by SWWM. One method is to construct the water balance equation from the ground hydrological station observation data and use the runoff depth as the standard to verify the model simulated runoff results, whose main advantage is to quantitatively verify the runoff volume simulated by SWMM. Another method is to detect surface flooding by passive microwave remote sensing and use the normalized difference frequency index (NDFI) to detect flooding; an NDFI is the detected presence of surface water sensitivity, which has the advantage of responding to the inter-annual variation of the flood peak over time.

#### 2.4.1. Water Balance for Calibration

Some sensitive model parameters were calibrated using data from a 2010 flood, as reported by Li et al. [39]. Considering the lack of flood discharge data for central Mianyang, the model parameters were calibrated based on the maximum discharge of the entire study

area in 2010, as inferred from a water balance calculation that included the surrounding tributaries and river branches. The basic concept of a water balance calculation is shown in Figure 3 [32]. The calculation steps are as follows:

$$R_{\text{est}} = \Delta Q \left( \frac{A_c}{\Delta A} \right) \tag{3}$$

$$\Delta Q = Q_3 - (Q_1 + Q_2 + Q_4) \tag{4}$$

$$\Delta A = A_3 - (A_1 + A_2 + A_4) \tag{5}$$

where $R_{\text{est}}$ is the runoff from the target area, to be estimated by water balance calculation; $A_c$ is area of the model domain, $Q_1$, $Q_2$, and $Q_3$ are the peak flow at the Fujiangqiao, Anchang, and Fenggu gauging stations, respectively. The peak flow from branch $Q_4$ is estimated by assuming similar specific discharge of this branch to that of the Furongxi gauging station ($Q_1$). Hence, $R_{est}$ is estimated peak flow as 189.6 m$^3$/s and, hereafter, this value is referred to as the "estimated runoff" and used as a reference for calibrating the model. The estimated runoff is also representing runoff generation from the study area, which is the central part of Mianyang city.

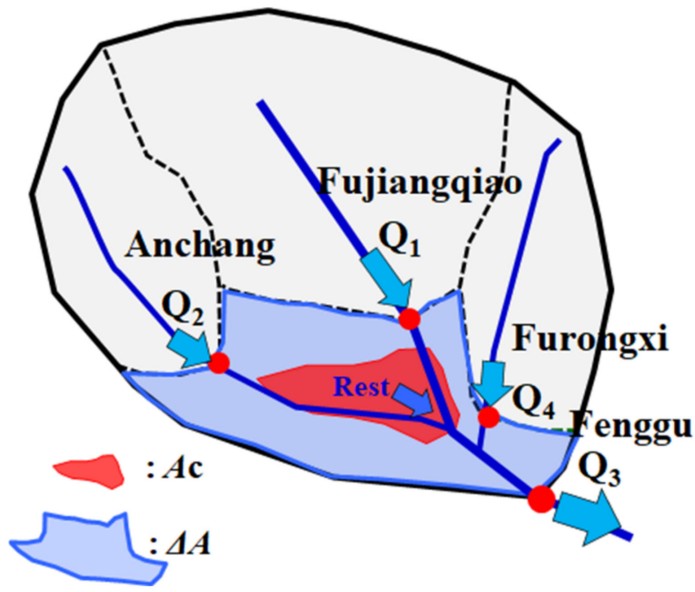

**Figure 3.** Schematics of water balance analysis [39].

2.4.2. Satellite Observations for Validation

Passive microwave remote sensing effectively detects surface moisture and flooding. The 19- and 22-GHz frequency with vertical polarization channels, respectively, of the dedicated special sensor microwave imager (SSM/I) are sensitive to surface moisture and flooding because they are minimally affected by cloud interference; we used these data from 1991 to 2020 (provided by the National Snow and Ice Data Center) as equal-volume cylindrical EASE-Grid projections with a resolution of 25 km, resampled to 0.25° (i.e., the grid size of our analysis). The Normalized Difference Frequency Index (NDFI) is used to detect surface moisture and flooding. *NDFI* is a sensitivity index for detecting the presence of surface water [41,42] and is expressed as follows:

$$\text{NDFI} = \frac{TB_{22V} - TB_{19V}}{TB_{22V} + TB_{19V}} \tag{6}$$

where $TB_{22V}$ and $TB_{19V}$ are the brightness temperatures with vertical polarization at 22 and 19 GHz, respectively. The maximum NDFI (*NDFIm*) at each pixel during summer was calculated for each year.

Flood magnitude was quantified based on the deviation (anomaly) of the *NDFIm* from the long-term average for 2015–2020. Anomalies were calculated as follows:

$$\text{a}NDFIm_{\text{year}} = \frac{NDFIm_{\text{year}}(x,y) - \mu_{NDFIm}(x,y)}{\sigma_{NDFIm}(x,y)} \tag{7}$$

where the subscript denotes the target year, x and y give the pixel location (longitude and latitude, respectively), $NDFIm_{\text{year}}$ (x, y) is the maximum *NDFI* over the June to September, and $\mu_{NDFIm}$ (x, y) and $\sigma_{NDFIm}$ (x, y) are the multi-year (1991–2020) average and standard deviation of *NFDIm* (x, y), respectively. As *aNDFIm* become more positive for a given year, *NDFIm* increase markedly in that year compared to other years. Higher-anomaly regions experienced extraordinarily intense or high-volume surface flooding. We have extracted the *NDFI* data from Li et al. [43].

## 2.5. Rainfall Observation Data and Design Rainfall Scenarios

Hourly rainfalls were obtained from the Fujiangqiao station of the Sichuan Water Business Platform; these data were combined with GSOD data to calculate the annual maximum daily rainfall intensities for different return periods, using the daily rainfall records from 1973 to 2017. The probability density function was normally distributed, and frequency analysis was thus performed using the Rainbow packing tool [44]. Next, three different downscaling methods were used to generate hourly rainfall time series for runoff analysis of different return periods.

### 2.5.1. Historical Patterns

The historical patterns are the maximum daily rainfalls for each month. The hourly rainfall time series of the eight maximum flood events from June to September 2015–2020 were separately analyzed; four were single-peak cases and the other four were multi-peak cases. Then, daily design rainfalls with different return period were temporally downscaled into hourly hyetograph following the temporal pattern of hourly rainfall record of each historical events.

### 2.5.2. Chi-Squared Probability Distribution Rainfall Patterns

The chi-squared probability distribution rainfall patterns were developed by Ye et al. [45] and applied to flood simulation for Mianyang by Li et al. [40]; they consider all rainfall fields in each month. The data were then input to the urban runoff simulation model. The temporal downscaling process is outlined in Figure 4. The temporal pattern of hourly rainfall is assumed to exhibit a chi-squared probability distribution (Equation (10)), and the peak intensity and duration of hourly rainfall are modeled as follows:

$$T = \alpha + \beta \ln(P) \tag{8}$$

$$P = a + b P_A \tag{9}$$

$$X^2(x : n) = \begin{cases} \dfrac{x^{0.5n-1} e^{-0.5x}}{2^{0.5n} \Gamma(0.5n)} , x > 0 \\ 0 , x \leq 0 \end{cases} \tag{10}$$

$$\Gamma(s) = \int_0^{+\infty} t^{s-1} e^{-t} \mathrm{d}t , s > 0 \tag{11}$$

where $T$ is the duration of precipitation (*h*), $P$ is the total daily precipitation (mm), $P_A$ is the maximum precipitation intensity (mm/h), and $\alpha$, $\beta$, and *a*, *b* are model parameters. Because the degrees of freedom (*n* values) of the output varies with $T$, the shape of the chi-square distribution also varies with $T$ through changes in *n*, as shown in Figure 4

and Table 2. Parameters $\alpha$, $\beta$, and $a$, $b$ were calculated for each month, based on the hourly precipitation data recorded at the meteorological station in Fujiangqiao during the period 2015–2020. With this downscaling method, $P_A$ and $T$ are estimated based on $P$ using Equations (8) and (9). The temporal pattern of hourly precipitation over time $T$ is based on the chi-squared distribution, but the peak intensity $P_A$ is not. Finally, the hourly precipitation time data (with the exception of $P_A$) were adjusted to ensure that the total precipitation over $T$ was equal to $P$.

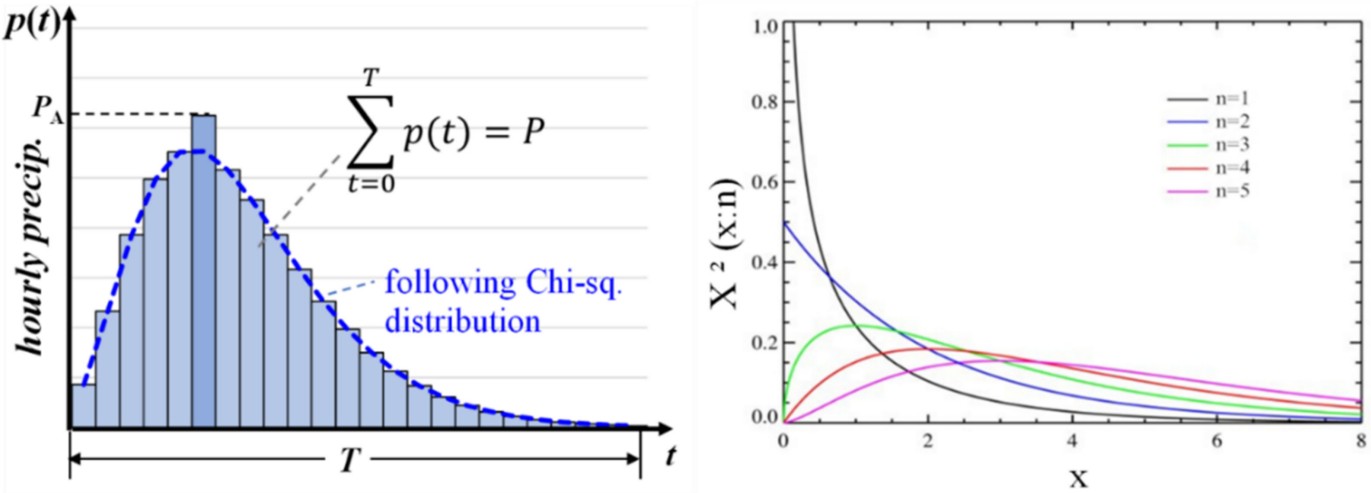

**Figure 4.** Chi-square distribution probability density function plot [33].

**Table 2.** The rainfall ephemeris corresponds to the degree of freedom [33].

| T [h] | (1, 8] * | (8, 11] | (11, 14] | (14, 16] | (16, 18] | (18, 24] |
|-------|----------|---------|----------|----------|----------|----------|
| n     | 3        | 4       | 5        | 6        | 7        | 8        |

* (1, 8]: $1 < T \leq 8$.

### 2.5.3. Chicago Design Storm

The Chicago design storm is widely used when modeling rainfall scenarios for Chinese sponge cities [46]; the rainfall pattern is determined by Equations (8) and (12). Considering the empirical storm equation provided by the local government of Mianyang, we employed the Chicago design storm as follows:

$$i = \frac{5.28\,(1 + 0.721 \log P)}{(t + 4.724)^{0.501}} \tag{12}$$

where: $i$ is the peak intensity of rainfall in mm/min, $t$ is the rainfall calendar time in min, and $P$ is the return period of daily rainfall in years.

First, the peak coefficients were defined as the ratios of the time of flood peaking to the total rainfall duration. Four single-peak rain types with different peak times were used, corresponding to rainfall peak coefficients of 0.2, 0.4, 0.6, and 0.8; six rainfall return periods of P = 1, 2, 5, 10, 20, 50, and 100 years were adopted. The rainfall pattens are shown in Figure 5 (as an example of 5 years return period). We used the same peak coefficient (0.5) for two multi-peak rain patterns (i.e., with two and three peaks); we employed six rainfall return periods. The process is shown in Figure 6 (as an example of 2 years return period). A similar pattern was used for temporal downscaling from daily design rainfalls with different return period into hourly hyetograph.

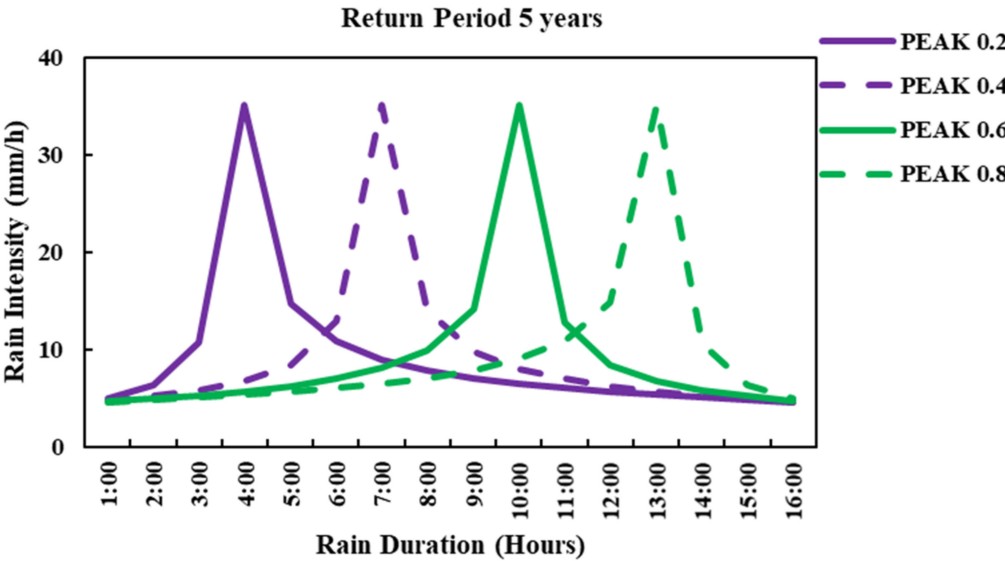

**Figure 5.** Single peak Chicago design storm by different peak coefficient.

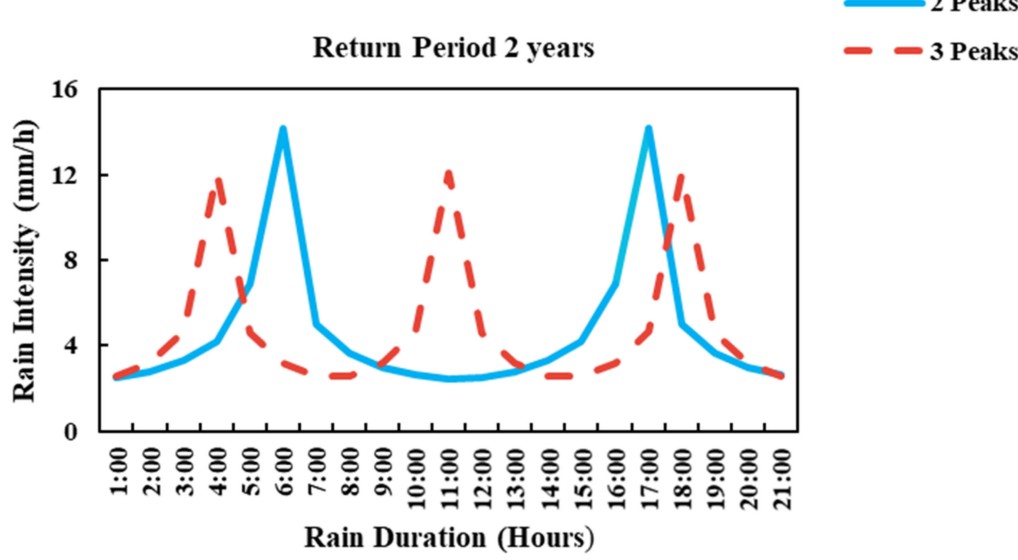

**Figure 6.** Multi-peak Chicago design storm by different peak coefficient.

### 2.6. Planning of LID Measures for Sponge City in Mianyang City

The Mianyang Sponge City Special Plan (2016) states that the following four LIDs will be installed:

(1) Green roofs (GRs): vegetated soil above drainage mats that serve to convey stormwater [47].
(2) Permeable pavement (PP): pavement of high porosity and permeability that allows some rainwater through [48].
(3) Rain gardens (RGs): water is retained in surface depressions filled with vegetated soil on a gravel storage bed [49].
(4) Rain barrels (RBs): water tanks are used to capture runoff, typically via pipes from rooftops [50].

The areas and percentages of each LID facility (for the entire study area) are shown in Table 3. The RB unit is $m^3$ but was converted to $m^2$ by setting the rain barrel height to 1.0 m. The parameters were derived from design sheets, SWMM manuals, and other literature [51–53].

**Table 3.** Type of LIDs and Coverage.

| Type of LIDs | Green Roofs (GRs) | Permeable Pavement (PP) | Rain Gardens (RGs) | Rain Barrels (RBs) |
|---|---|---|---|---|
| Description |  |  |  |  |
| Area (km²) | 9.95 | 24.50 | 27.12 | 3.11 |
| Ratio (%) | 4.76 | 11.71 | 12.96 | 1.49 |

*2.7. Experimental Design*

Rainfall may exhibit one or more peaks. The single-peak falls are divided into two groups that differ in terms of intensities and peak coefficients; the multi-peak falls are divided into two groups that differ in terms of rainfall intensities and peak numbers. The groupings are shown in Table 4; the design precipitation levels for 2, 5, 10, 20, 50, and 100-year return periods were considered first. Each experiment was repeated 24 times based on the different return periods for the months of June to September inclusive.

**Table 4.** Experimental design.

| Experiments | Peak Types | Number of Peaks | Peak Coefficients | Methods |
|---|---|---|---|---|
| E1 | Single | 1 | 0.3–0.7 | **His** |
| E2 | Single | 1 | 0.2–0.3 | **Chi-2** |
| E3 | Single | 1 | **0.2** | Chicago |
| E4 | Single | 1 | **0.4** | Chicago |
| E5 | Single | 1 | **0.6** | Chicago |
| E6 | Single | 1 | **0.8** | Chicago |
| E7 | Multi | **2–4** | 0.2–1 | His |
| E8 | Multi | **2** | 0.5 | Chicago |
| E9 | Multi | **3** | 0.3 | Chicago |

The specific description of the experimental group is as follows:

(1) Single-peak Extreme rainfall (E1–E2). The E1 single peak historical patterns served as the June-to-September single-peak extreme rainfall scenario. In E2, the chi-squared probability distribution of single-peak rainfall pattern was employed; this is the June-to-September average rainfall.

(2) Single-peak Peak coefficients (E3–E6). In E3–E6, the Chicago design storm single-peak rainfall patterns created by weather generator [39] were used; these are the flood peaks with coefficients of 0.2, 0.4, 0.6, and 0.8 from June to September.

(3) Multi-peak (E7–E9). In E7, an historical multi-peaked rainfall rain pattern was used; this is the June-to-September multi-peak extreme rainfall scenario. In E8, the Chicago design storm multi-peak rainfall pattern created by the weather generator was used to represent the June-to-September average double-peak rainfall pattern when the average number of peaks was 2. In E9, the Chicago design storm multi-peak rainfall rain pattern created by the weather generator was also used; the mean peak number was 3 for June-to-September.

In all experiments, urban runoff simulations were performed when LID practices were and were not implemented. The effects of LIDs on flood control were assessed by calculating the "reduction rates" of flood peak and volume; each rate is the difference in flood peak or volume between the presence and absence of the LID initiatives.

## 3. Results and Discussion

### 3.1. Validation

#### 3.1.1. Water Balance

Model reliability was quantitatively assessed using the runoff depth derived via water balancing, although the simulated annual maximum discharge Rcal of the model using the default parameters (156.7 m$^3$/s) was underestimated by 17% compared to the estimated runoff. However, the model reasonably yielded discharge outputs from rainfall inputs. To improve model performance, a (sensitive) parameter (the depth of depression storage in impervious areas) and the infiltration parameters were re-evaluated (i.e., calibrated) to reduce the difference between Rcal and Rest. The sensitive parameters were identified via a literature review [54] and used for a preliminary model simulation (parameter sensitivity analysis). Rcal became 164.9 m$^3$/s (a 13% underestimate) after parameter calibration.

#### 3.1.2. Satellite Observations

Simulation reasonably represented both the flood peak for a specific flood event (in 2010, validated by the water balance) and the differences in flood magnitudes. After sensitive parameters had been determined using the water balance equation, remote sensing data were introduced to further evaluate accuracy and reliability. The results are shown in Figure 7. When comparing the annual maximum flood events (aNDFImyear) and the simulated discharge of the no-LID SWMM CASE in 2015–2020, the inter-annual variation correlation coefficient was 0.6. The reason for the difference in 2015 and 2018 was that the spatial resolution of satellite data was larger than the study area. However, we can still conclude that they were strongly correlated [55]; the SWMM simulations were consistent with the satellite data.

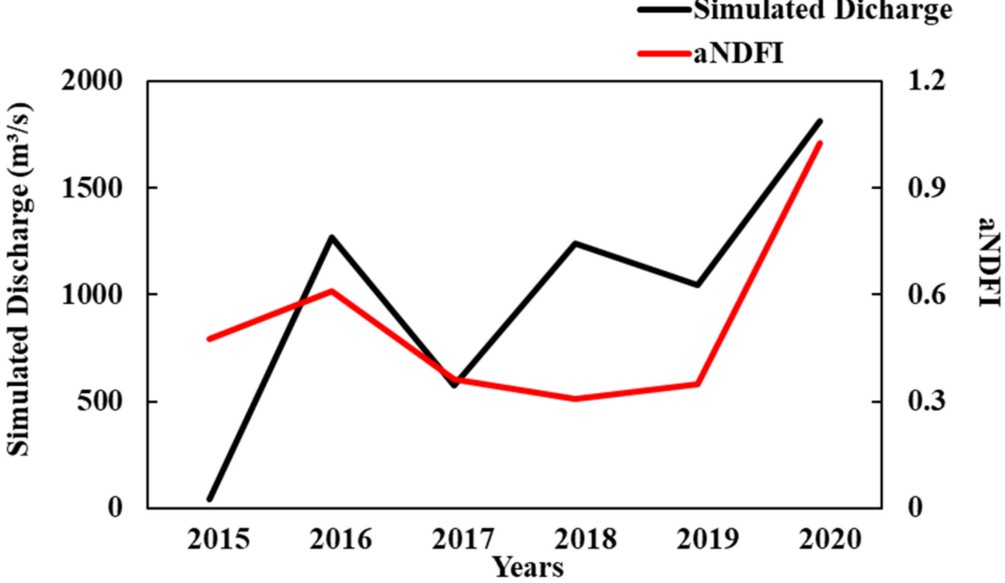

**Figure 7.** Annual maximum flood events aNDFIm$_{year}$ and simulated discharge.

### 3.2. Effect of Single Peak

#### 3.2.1. Extreme and Average Conditions

Rainfall Patterns Analysis

Both E1 and E2 have return periods of 2–100 years and consider the June-to-September single-peak rainfalls; the rainfalls were equal except in July (where the rainfall duration was less in E1 than in E2, indicating that the average rainfall intensity was greater in E1 than in E2). The principal reason for the difference between July and other months is that the rainfall duration of the field representing the extremes in July is long and the rainfall distribution is extremely uneven. However, the flood rainfall intensity is generally greater

in E1 than in E2, which is consistent with the concept that E1 represents monthly extreme rainfall and E2 represents monthly average rainfall. Notably, the flood rainfall intensity is slightly greater in E2 than in E1 during the 2–20-year return periods because of the generally long duration and low intensity of rainfall in September. The PEAK BIAS was defined as the difference in rainfall peak intensities. The average BIAS was the experimental difference in average rainfall intensities. The PEAK BIAS values of E2 compared to E1 in June-to-September were minus 33–34%, minus 65–66%, minus 36–38%, and plus 5–9%, respectively. The average BIAS values were minus 50–59%, plus 28–53%, minus 38–47%, and minus 53–59% in June-to-September, respectively (Figure 8, Tables 5 and 6).

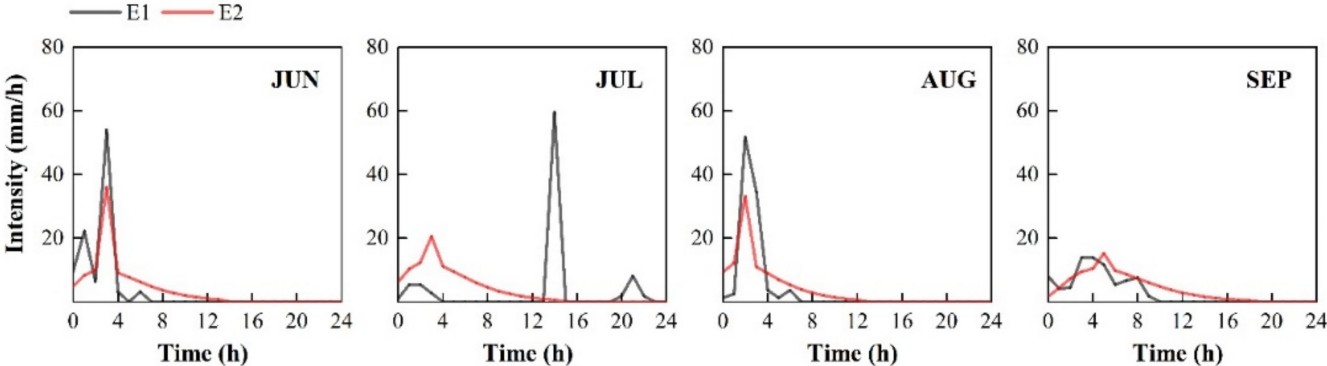

**Figure 8.** Extreme and average rainfall patterns analysis.

**Table 5.** PEAK BIAS and average BIAS for E1 and E2.

| Months | JUN | JUL | AUG | SEP |
|---|---|---|---|---|
| PEAK-BIAS (%) | [−34, −33] | [−66, −65] | [−38, −36] | [5, 9] |
| Average-BIAS (%) | [−59, −50] | [28, 53] | [−47, −38] | [−59, −53] |

**Table 6.** Extreme and average rainfall patterns analysis.

| Experiments | Month | Return Period (y) | Duration (h) | Peak Intensity (mm/h) | Average Intensity (mm/h) |
|---|---|---|---|---|---|
| E1 | JUN | 2–100 | 7 | 54.0–140.9 | 14.1–36.7 |
| | JUL | 2–100 | 23 | 59.6–155.6 | 4.3–11.2 |
| | AUG | 2–100 | 8 | 51.7–134.9 | 12.3–32.1 |
| | SEP | 2–100 | 9 | 13.9–36.2 | 10.9–28.6 |
| E2 | JUN | 2–100 | 14–17 | 35.9–92.6 | 7.0–15.1 |
| | JUL | 2–100 | 15–18 | 20.6–48.2 | 6.6–14.3 |
| | AUG | 2–100 | 13–15 | 33.2–83.6 | 7.6–17.1 |
| | SEP | 2–100 | 19–22 | 15.2–38.0 | 5.2–11.7 |

Flood Control Analysis

The experimental peak reduction rate was the peak difference between the LID and no-LID cases divided by the peak of the no-LID case; the volume reduction rate was the total volume difference between the LID and no-LID case divided by the total volume of the no-LID case. For E1, the peak and volume reduction rates exhibited decreasing trends in all months as the return period increased. Thus, under the rain pattern of monthly extreme rainfall, the LID facilities became saturated within the 2-year return period. As rainfall increased, the LID facilities could not manage the excess. Importantly, September exhibits a long rainfall duration but a low flood rain intensity; accordingly, the peak reduction rate in September increased slowly during the 2–20-year return period and decreased sharply thereafter, indicating that the LID facilities were saturated during only the 20–50-year return period under September rain pattern conditions. Because the average September

rain intensity is also low, the decreasing trend of the volume reduction rate is also smaller than the decreasing trends in other months.

For E2, as the return period increased, the peak and volume reduction rates tended to first increase and then decrease in each month. These findings indicate that, under the chi-squared rain pattern, the return periods at which the LID facilities became saturated varied according to the month. The critical point peak reduction rates of June, July, August, and September were 10–20 years, 50–100 years, 10–20 years, and 50–100 years, respectively. June and August are similar; the flood peaks and average rainfall intensities are much higher in these months than in other months. July and September are similar; the flood peaks and average rainfall intensities are much lower in these months than in other months. Thus, regardless of similar total rainfall, greater peak and average rainfall intensities are associated with an earlier return period at which the LID facilities become saturated.

The peak E1 reduction rate is 13.8–31.3% and the volume reduction rate is 22.6–30.7%; the peak E2 reduction rate is 25.6–30.9% and the volume reduction rate is 28.7–31.2%. The E1 figures are smaller than the E2 figures, indicating that urban runoff accumulation caused by natural extreme rainfall is greater under average rainfall conditions, while the response of LID facilities is weaker (Figure 9).

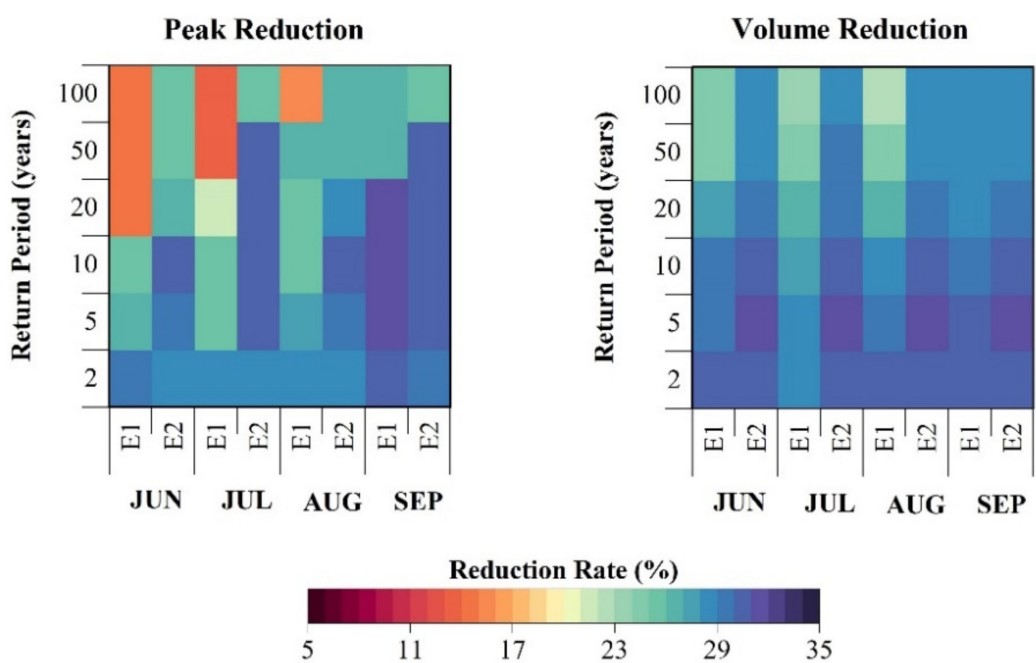

**Figure 9.** Extreme and average flood control analysis.

3.2.2. Different Peak Timing
Rainfall Patterns Analysis

E3–E6 are all single-peak rainfall types with equal total rainfall from June to September of the 2–100-year return periods; the only differences are in peak timing (peak coefficients of 0.2, 0.4, 0.6, and 0.8, respectively). Thus, the intensity of the Chicago rainfall pattern is lower than the intensity of the natural extreme rainfall pattern. Compared to E1, the PEAK BIAS values of E3–E6 were minus 56–62%, minus 60–65%, minus 52–60%, and plus 30–50% in June-to-September, respectively. The average BIAS values were minus 50–59%, 28–53%, minus 38–47%, and minus 53–59% (Figure 10, Tables 7 and 8).

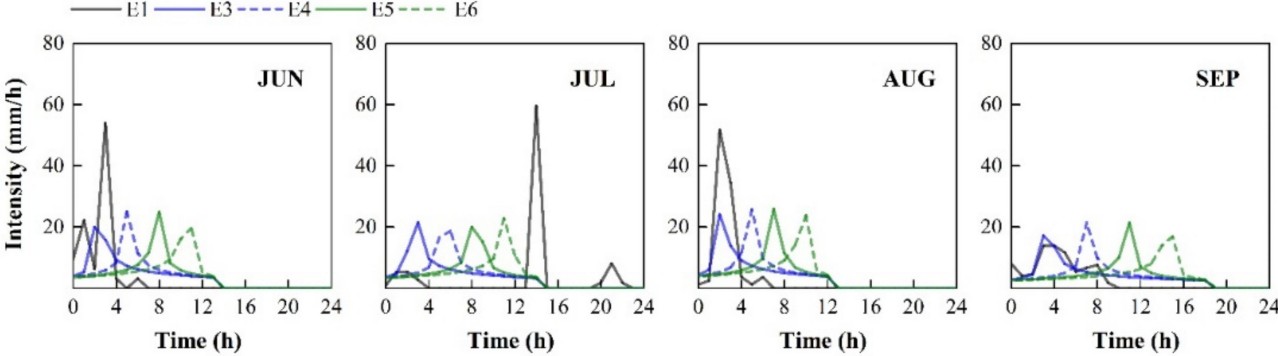

**Figure 10.** Different peak timing rainfall patterns analysis.

**Table 7.** PEAK BIAS and Average BIAS for E1, E3, E4, E5, and E6.

| Months | JUN | JUL | AUG | SEP |
|---|---|---|---|---|
| PEAK-BIAS (%) | [−62, −56] | [−65, −60] | [−52, −60] | [30, 50] |
| Average-BIAS (%) | [−59, −50] | [28, 53] | [−47, −38] | [−59, −53] |

**Table 8.** Different peak timing rainfall patterns analysis.

| Experiments | Month | Return Period (y) | Duration (h) | Peak Intensity (mm/h) | Average Intensity (mm/h) |
|---|---|---|---|---|---|
| E3–E6 | JUN | 2–100 | 14–17 | 22.4–56.7 | 7.0–15.1 |
| | JUL | 2–100 | 15–18 | 20.7–55.1 | 6.6–14.3 |
| | AUG | 2–100 | 13–15 | 24.8–54.2 | 7.6–17.1 |
| | SEP | 2–100 | 19–22 | 19.2–49.8 | 5.2–11.7 |

Flood Control Analysis

Considering the lags in peak time, the peak reduction rates of E3–E6 all exhibit decreasing trends for return periods greater than 10 years, while the volume reduction rate is not significant. Thus, at the longer times, the LIDs are saturated at a peak coefficient of 0.2; with increasing lag in flood time, the ability of LID facilities to cope becomes increasingly weaker. The peak reduction rates of E3–E6 are 24.4–30.7% and the volume reduction rates are 28.0–31.3%. As the return period increases, the reduction rate of E1 becomes smaller than the reduction rate of E3–E6. Thus, a larger return period (relative to the Chicago rainfall type conditions) is associated with a larger difference between urban runoff accumulation caused by natural extreme rainfall and accumulation caused by the Chicago rainfall type, as well as a weaker response of the LID facilities (Figure 11).

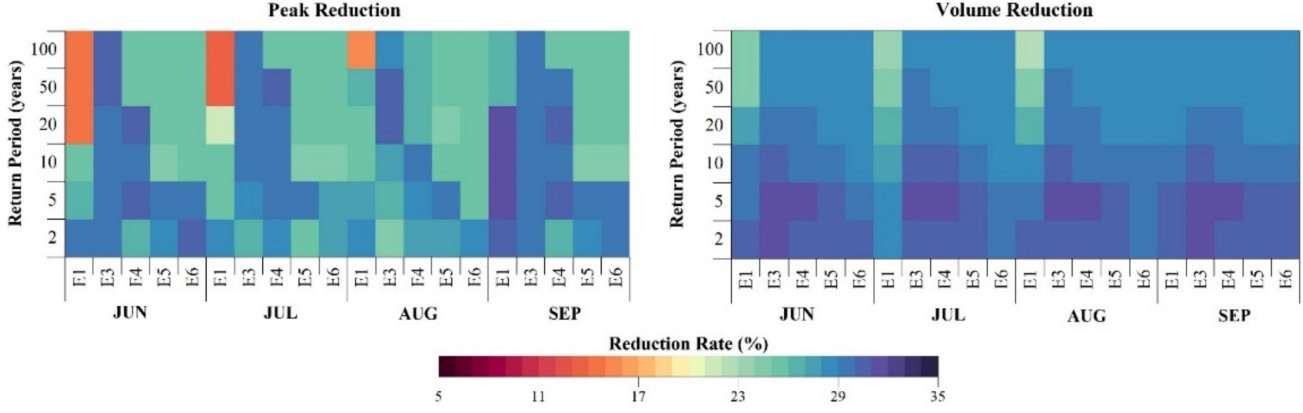

**Figure 11.** Different peak timing flood control analysis.

### 3.3. Effect of Multi Peak

#### 3.3.1. Rainfall Patterns Analysis

E7, E8, and E9 are 2–100-year return period June-to-September multi-peak rainfall patterns with equal total rainfall. The rainfall frequency and average rain intensity are similar for each month; E7–E9 differ in peak numbers and flood rain intensities. E8 is the uniform double-peaked rainfall of the Chicago rain pattern, while E9 is the uniform multi-peaked rainfall of the Chicago rain pattern. The peak intensity is E7 > E8 > E9. In addition, the PEAK BIAS values of E8 over E7 were minus 45%, minus 51%, minus 47%, and minus 7% in June to September, respectively. The PEAK BIAS values of E9 over E7 were minus 57%, minus 58%, minus 43%, and minus 27% in those months (Figure 12, Tables 9 and 10).

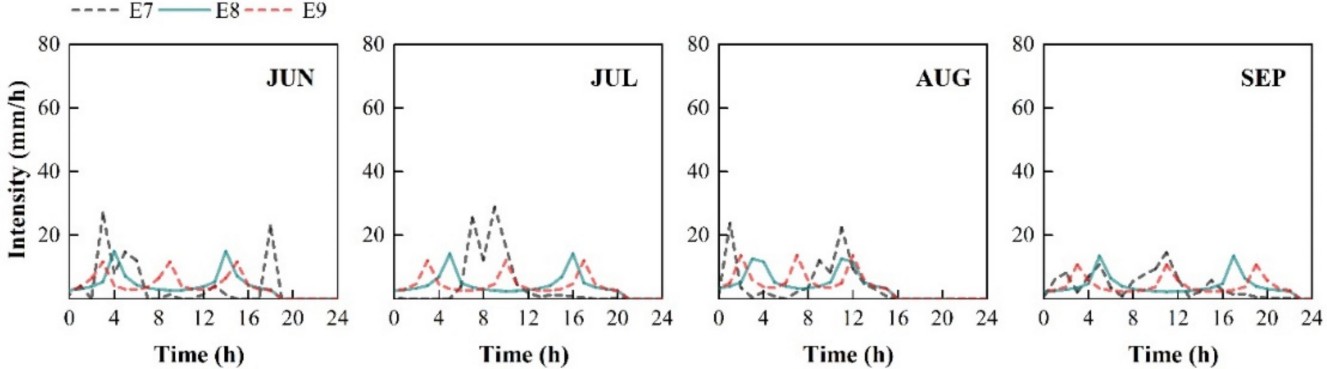

**Figure 12.** Multi peak rainfall patterns analysis.

**Table 9.** PEAK BIAS for E7, E8, and E9.

| PEAK-BIAS | JUN | JUL | AUG | SEP |
|---|---|---|---|---|
| E7 & E8 | −45 | −51 | −47 | −7 |
| E7 & E9 | −57 | −58 | −43 | −27 |

**Table 10.** Multi peak rainfall patterns analysis.

| Experiments | Month | Return Period (y) | Duration (h) | Peak Intensity (mm/h) | Average Intensity (mm/h) |
|---|---|---|---|---|---|
| E7 | JUN | 2–100 | 19 | 27.0–70.4 | 5.2–13.5 |
|  | JUL | 2–100 | 21 | 29.0–75.7 | 4.7–12.2 |
|  | AUG | 2–100 | 16 | 23.7–62.0 | 6.2–16.1 |
|  | SEP | 2–100 | 23 | 14.6–38.1 | 4.3–11.2 |
| E8 | JUN | 2–100 | 19 | 15.0–39.1 | 5.2–13.5 |
|  | JUL | 2–100 | 21 | 14.1–36.9 | 4.7–12.2 |
|  | AUG | 2–100 | 16 | 12.6–33.0 | 6.2–16.1 |
|  | SEP | 2–100 | 23 | 13.6–35.5 | 4.3–11.2 |
| E9 | JUN | 2–100 | 19 | 11.7–30.5 | 5.2–13.5 |
|  | JUL | 2–100 | 21 | 12.1–31.5 | 4.7–12.2 |
|  | AUG | 2–100 | 16 | 13.6–35.6 | 6.2–16.1 |
|  | SEP | 2–100 | 23 | 10.7–28.0 | 4.3–11.2 |

#### 3.3.2. Flood Control Analysis

For E7, the peak reduction rate increases with increasing return periods in June and August; thus, when peaks are more than 10 h apart, they exhibit minimal interaction and the LID facilities are not saturated. In July, the peak reduction rate begins to decrease at return periods of 5–10 years because the July flood peaks are only 1 h apart and the peak rain intensity is highest in that month. In September, the peak reduction rate also begins to

decrease after return periods of 5–10 years; this reduction is less than in July because there are more continuous flood peaks in September. For E8, the peak reduction and volume reduction rates first increase, then decrease; the LID saturation thresholds in June and September both occur at return periods of 10–20 years. The volume reduction rates differ slightly; the July threshold has a return period of 10–20 years, while the other months have return periods of 5–10 years. For E9, the saturation thresholds for the LID facilities at the reduced peak rates all occurred in the 50–100-year return periods, while the saturation thresholds for LID facilities of reduced capacity occurred at the 5–10-year return periods.

The peak E7 reduction rate is 14.5–30.4% and the volume reduction rate is 26.1–30.9%; the peak E8 reduction rate is 21.6–31.0% and the volume reduction rate is 27.0–31.0%; the peak E9 reduction rate is 19.9–29.8% and the volume reduction rate is 24.9–30.9%. A comparison of E8 and E9 revealed that a higher number of wave peaks was associated with lower flood rainfall intensity, as well as smaller peak and volume reduction rates. The difference in urban runoff caused by natural extreme rainfall and Chicago rain type uniform multi-peak rainfall under multi-peak conditions is not substantial, except under particularly extreme conditions, such as when flood peaks are very close in July and the flood rain intensity is maximal, with a return period of 100 years (Figure 13).

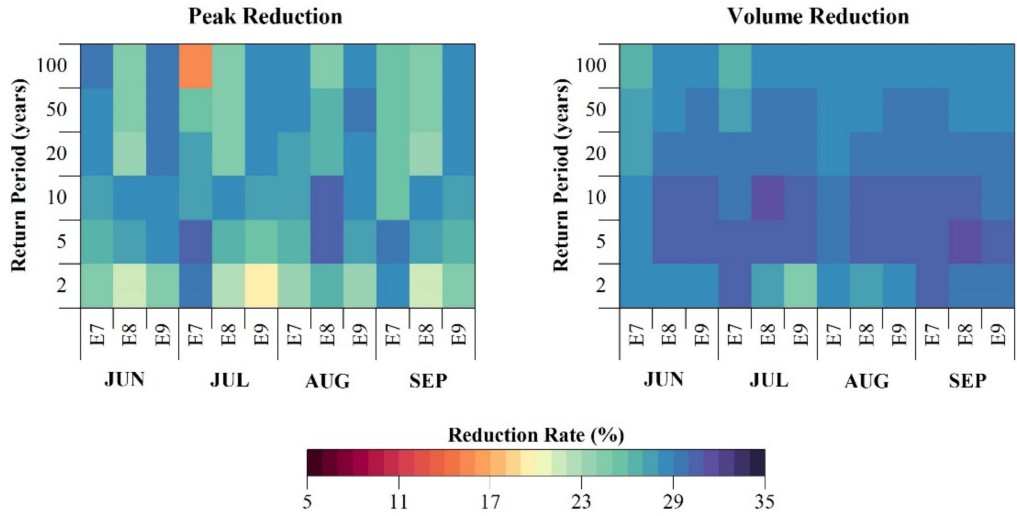

**Figure 13.** Effect of multi peak flood control analysis.

## 4. Conclusions

We studied the Mianyang Sponge City of Sichuan Province. We used different return periods and rain types (historical, chi-squared, and Chicago rain types) to evaluate the effects of LID measures on peak reduction from both single- and multi-peak perspectives; we validated our model using water balance and remote sensing techniques. Our principal conclusions were:

(1) The model underestimates hourly runoff over large areas by approximately 13%, as verified by water balancing and remote sensing. The simulated runoff trend was strongly correlated with the satellite observations.

(2) The flood peak and mean rainfall intensities were generally larger for single-peak historical rainfalls than for the chi-squared rain pattern; the difference in bias was substantial, except for the peak bias in September (long continuous rainfall). The peak and average rainfall intensities were also generally lower for the single-peak Chicago rainfall type than for the single-peak historical rainfall; the peak and average biases were equally large. The multi-peak historical rainfall pattern was identical to the multi-peak Chicago pattern; however, the flood rainfall intensity was generally larger in the multi-peak historical pattern than in the multi-peak Chicago rainfall pattern.

(3) Simulation revealed that the ability of LID facilities to control flood peaks and volumes was weaker under the single-peak chi-squared rainfall pattern than under the historical rainfall pattern. Control became weaker as the flood peaks became closer. For multi-peak rainfall, the difference in urban runoff caused by natural extreme rainfall and the uniform multi-peak rainfall of the Chicago rain type was not substantial, while the ability of LID facilities to control flood peaks and volumes became progressively weaker as the average wave peak increased.

In contrast to the natural extreme rainfall rain patterns, artificial rain patterns overestimate the ability of LID facilities to control flood peaks and flood volumes. During sponge city design, LID facilities should be optimally placed with reference to local topography and both spatial and temporal rainfall characteristics to ensure effective flood control.

A limitation of this study is that we could not deal with two-dimensional flood inundation processes. Use of a two-dimensional hydrological model is necessary to simulate the impact of LIDs on spatio-temporal distribution of inundation extent in urban areas. In addition, parameter calibration of SWMM still needs to be improved and validation (accuracy check) using new data will be one of the future tasks. The impact of LIDs on urban runoff was evaluated in this study; however, PPs and RGs would also have effects on groundwater through the change of infiltration and soil water storage. Quantifying the impact of PPs and RGs on groundwater level will be discussed in future works.

**Author Contributions:** The manuscript was primarily written by H.L., with H.I. and Y.W. contributing to its preparation. H.I., K.S. and J.M. supervised the research and critically reviewed the draft. All authors have read and agreed to the published version of the manuscript.

**Funding:** This research was supported by Interdisciplinary Centre for River Basin Environment—University of Yamanashi, and the Support for Pioneering Research Initiated by the Next Generation—the Japan Science and Technology Agency.

**Data Availability Statement:** Not applicable.

**Acknowledgments:** We would like to thank Mianyang Water Resources Bureau and Mianyang Natural Resources Bureau for providing the sponge city planning of Mianyang and hydrological data, the National Geomatic Center of China (NGCC) for providing the GLOBELAND 30 data, and the National Snow & Ice Data Center (NSIDC) for providing the SSM/I data.

**Conflicts of Interest:** The authors declare no conflict of interest.

## Abbreviations

| | |
|---|---|
| LID | Low-impact development |
| SWMM | Storm Water Management Model |
| SHE | Système Hydrologique Européen |
| SWAT | Soil & Water Assessment Tool |
| IHDM | Institute of Hydrology Distributed Model |
| Chi-2 | Chis-quared probability distribution rainfall patterns |
| Chicago | Chicago design storm |
| His | Historical patterns |
| GRs | Green roofs |
| PP | Permeable pavement |
| RGs | Rain gardens |
| RBs | Rain barrels |
| GSSD | Global Surface Summary of the Day |
| ASTER-GDEM | Advanced Spaceborne Thermal Emission and Reflection Radiometer |
| SSM/I | Special Sensor Microwave/Imager |
| NDFI | Normalized difference frequency index |

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
