# Peer review of "Assessment of Sponge City Flood Control Capacity According to Rainfall Pattern Using a Numerical Model after Muti-Source Validation"

_water, doi:10.3390/w14050769_

Round 1

Reviewer 1 Report

In general, the article is properly prepared. In the following, my comments and suggestion are presented.

** At the end of the abstract, add a sentence about the application of your finding for water or urban managers.  

** Lines 36, 37, and 40: Change the currency from RMB to USA Dollar. 

** Lines 51-52: add a citation for each model.

** Line 152: In the study area section, add a geographical direction+altitude of your studied city, and same for study site.

** Conclusion: Two parts are necessary to add here: research limitation(s) and research direction for future scientists. 

Author Response

Response to Reviewer 1 Comments:

Dear reviewer, We would like to thank you for the helpful comments provided for our paper. Please find our response to your comments.

1.At the end of the abstract, add a sentence about the application of your finding for water or urban managers.

Response: We have added as follow (L23-25):

In this context, the results obtained in this study provide useful reference information about impact of rainfall patterns on urban flood control by LID, and can be used for sponge city design in other parts of China.

2.Lines 36, 37, and 40: Change the currency from RMB to USA Dollar.

Response: We have revised based on reviewer’s suggestion and changed as follow (L36-42):

In 2012, heavy rain on 20 July in Beijing caused 10,660 houses to collapse; 1.602 million people were affected and the direct economic loss exceeded $ 1.84 billion [5]. In 2016, a rainstorm on 6 July in Wuhan affected 757,000 people and caused direct economic loss-es of $ 36 million [6]. In 2020, heavy rain on 22 May in Guangzhou caused suspension of the subway and great economic losses [7]. In July 2021, Zhengzhou (Henan) was affected by a severe rainstorm that killed 51 people and caused direct economic loss of $ 10.4 billion [8].

3.Lines 51-52: add a citation for each model.

Response: We have added some literature to the references and have cited it within the revised text (L53-55, L599-606).

SWMM:

[12] Mobilia, M., & Longobardi, A. (2020). Impact of rainfall properties on the performance of hydrological models for green roofs simulation. Water Science and Technology, 81(7), 1375-1387, doi:10.2166/wst.2020.210

SHE:

[13] Rujner, H., Leonhardt, G., Marsalek, J., & Viklander, M. (2018). High-resolution modeling of the grass swale response to runoff inflows with Mike SHE. Journal of Hydrology, Volume 562, 411-422, doi:10.1016/j.jhydrol.2018.05.024

SWAT:

[14] Seo, M., Jaber, F., Srinivasan, R., & Jeong, J. (2017). Evaluating the Impact of Low Impact Development (LID) Practices on Water Quantity and Quality under Different Development Designs Using SWAT. Water, 9(3), 193, doi:10.3390/w9030193

IHDM:

[15] Beven, K J, Calver, A and Morris, E M. (1987). The Institute of Hydrology Distributed Model, Institute of Hydrology Report 98, Wallingford, UK.

4.Line 152: In the study area section, add a geographical direction+altitude of your studied city, and same for study site.

Response: We have added as follow (L158-159):

The topography is high in the north and low in the south, high in the east and west and low in the middle, altitude 450-538m.

5.Conclusion: Two parts are necessary to add here: research limitation(s) and research direction for future scientists.

Response: We thank the reviewer for this important comment, and added as follow (L536-543):

A limitation of this study is that we could not deal with 2-dimensional flood inundation processes. Use of a 2-dimensional hydrological model is necessary to simulate impact of LIDs on Spatio-temporal distribution of inundation extent in urban areas. In addition, parameter calibration of SWMM still needs to be improved and validation (accuracy check) using new data will be one of the future tasks. The impact of LIDs on urban runoff was evaluated in this study, however, PPs and RGs would also have effects on groundwater through the change of infiltration and soil water storage. Quantifying the impact of PPs and RGs on groundwater level will be discussed in future works.

Author Response

Response to Reviewer 2 Comments:

Dear reviewer, We would like to thank you for the helpful comments provided for our paper. Please find our response to your comments.

  1. A section with acronyms is required

Response: Thanks to the reviewer for the suggestion. We have added section of Abbreviations (acronyms) as follow (L557-574):

Abbreviations

LID Low-impact development

SWMM Storm Water Management Model

SHE Système Hydrologique Européen

SWAT Soil & Water Assessment Tool

IHDM Institute of Hydrology Distributed Model

Chi-2 Chi-squared probability distribution rainfall patterns

Chicago Chicago design storm

His Historical patterns

GRs Green roofs

PP Permeable pavement

RGs Rain gardens

RBs Rain barrels

GSSD Global Surface Summary of the Day

ASTER-GDEM Advanced Spaceborne Thermal Emission and Reflection Radiometer

SSM/I Special Sensor Microwave/Imager

NDFI Normalized difference frequency index

2.Please prefer the use of "the authors" instead of "we"

Response: We have revised based on reviewer’s suggestion (L552)

3.Lines 51 and 52: references are missing please see:

Response: Thanks to the reviewer for this very important suggestion. We have added some literature to the references and have cited it within the revised text.

SWMM:

[12] Mobilia, M., & Longobardi, A. (2020). Impact of rainfall properties on the performance of hydrological models for green roofs simulation. Water Science and Technology, 81(7), 1375-1387, doi:10.2166/wst.2020.210

SHE:

[13] Rujner, H., Leonhardt, G., Marsalek, J., & Viklander, M. (2018). High-resolution modeling of the grass swale response to runoff inflows with Mike SHE. Journal of Hydrology, Volume 562, 411-422, doi:10.1016/j.jhydrol.2018.05.024

SWAT:

[14] Seo, M., Jaber, F., Srinivasan, R., & Jeong, J. (2017). Evaluating the Impact of Low Impact Development (LID) Practices on Water Quantity and Quality under Different Development Designs Using SWAT. Water, 9(3), 193, doi:10.3390/w9030193

IHDM:

[15] Beven, K J, Calver, A and Morris, E M. (1987). The Institute of Hydrology Distributed Model, Institute of Hydrology Report 98, Wallingford, UK.

Reviewer 3 Report

1. Although Li et al. used the data in 2010 to calibrate the SWMM parameters (p.213-p.214), the difference between the water balance equation and the simulation value was 17%, so the sensitive parameters were corrected. It is suggested that the author can verify the revised parameters with the new data in the next few years. (p.362-p.371)

2. This study uses a 2-100-year return period, but only the rain patterns with different peak coefficients are explained in Figures 5 and 6. The return period is not shown in the figures. Please include the relevant information (p.305- p.306).

3. Mianyang City has planned four types of LID, among which PP and RGs are susceptible to groundwater rise and affect the functions of water storage and infiltration (p.314-p.324). It is suggested that the author can supply the groundwater level data during the rainstorm period in Mianyang to ensure LID will not be affected by the groundwater level.

4. The correlation coefficient between the simulated discharge and NDFImyear is 0.6. There is no standard for this evaluation, so it is difficult to prove the reliability of the model. Please supplement the evaluation criteria for the model (p.378-p.379).

5. The SWMM simulation and satellite data of 2015 and 2018 in Figure 7 contain discrepancies. It is necessary to explain the reason for the discrepancies or re-calibrate the parameters of SWMM (p.382).

6. E3 to E6 of the paper mentioned that the peak reduction rate and runoff reduction rate have a downward trend in the rain pattern of the 10-year return period. But the peak reduction rate of E3 in Figure 11 did not show a significant downward trend after the 10-year return period (p.451-p.452).

Author Response

Response to Reviewer 3 Comments:

Dear reviewer, We would like to thank you for the helpful comments provided for our paper. Please find our response to your comments.

1.Although Li et al. used the data in 2010 to calibrate the SWMM parameters (p.213-p.214), the difference between the water balance equation and the simulation value was 17%, so the sensitive parameters were corrected. It is suggested that the author can verify the revised parameters with the new data in the next few years. (p.362-p.371)

Response:

We thank the reviewer for this very important suggestion. Unfortunately, we could not access to detailed data on flood peak discharge after 2010. For the water balance calculation, flood peak discharge at all three gauging stations Q1, Q2 and Q3 should be necessary, however, these data are really available. So, validation (accuracy check) with the new data would be one of the future tasks, and we have added the following text. (L538-540)

In addition, parameter calibration of SWMM still needs to be improved and validation (accuracy check) using new data will be one of the future tasks.

  1. This study uses a 2-100-year return period, but only the rain patterns with different peak coefficients are explained in Figures 5 and 6. The return period is not shown in the figures. Please include the relevant information (p.305- p.306).

Response: We added the return period for Figures 5 and 6. And revised as follow (L308-315):

The rainfall pattens are shown in Figure 5 (as an example of 5 years return period). We used the same peak coefficient (0.5) for two multi-peak rain patterns (i.e., with two and three peaks); we employed six rainfall return periods. The process is shown in Figure 6 (as an example of 2 years return period).

  1. Mianyang City has planned four types of LID, among which PP and RGs are susceptible to groundwater rise and affect the functions of water storage and infiltration (p.314-p.324). It is suggested that the author can supply the groundwater level data during the rainstorm period in Mianyang to ensure LID will not be affected by the groundwater level.

Response:

We thank the reviewer for this very important suggestion. Impact of PP and RGs on groundwater would be very interesting topic and we really hope to investigate it in our future work. However, we could not access to groundwater level data in the study area, and could not supply groundwater level data during the rainstorm period. We have added the following text in the conclusion part as one of the future tasks (L540-543).

The impact of LIDs on urban runoff was evaluated in this study, however, PPs and RGs would also have effects on groundwater through the change of infiltration and soil water storage. Quantifying the impact of PPs and RGs on groundwater level will be discussed in the future works.

  1. The correlation coefficient between the simulated discharge and NDFImyear is 0.6. There is no standard for this evaluation, so it is difficult to prove the reliability of the model. Please supplement the evaluation criteria for the model (p.378-p.379).

Response: Our evaluation criteria, based on the interpretation of the correlation coefficient set by Concepts and Controversies 9th Edition, are shown below:

Correlation coefficient

0.8-1.0 Very strong correlation

0.6-0.8 Strongly correlated

0.4-0.6 Moderate correlation

0.2-0.4 Weak correlation

0.0-0.2 Very weak correlation or no correlation

We have added a book to the references and have cited it within the revised text.

[48] David S. Moore, William I. Notz. (2016) Statistics:Concepts and Controversies 9th Edition. W. H. Freeman.

  1. The SWMM simulation and satellite data of 2015 and 2018 in Figure 7 contain discrepancies. It is necessary to explain the reason for the discrepancies or re-calibrate the parameters of SWMM (p.382).

Response: Thank you for the comment. We checked the results again, and the reason for the difference in 2015 and 2018 is the inconsistency between the maximum flood times from the simulation and the maximum flood times from aNDFI. In this regard, we believe that higher resolution satellite data should be used in the future validation process. We have revised as follow (L379-384):

When comparing the annual maximum flood events aNDFImyear and the simulated discharge of the no-LID SWMM CASE in 2015-2020, the inter-annual variation correlation coefficient was 0.6, the reason for the difference in 2015 and 2018 is that the spatial resolution of satellite data is larger than the study area. but we can still conclude that they are strongly correlated [48]. the SWMM simulations were consistent with the satellite data.

  1. E3 to E6 of the paper mentioned that the peak reduction rate and runoff reduction rate have a downward trend in the rain pattern of the 10-year return period. But the peak reduction rate of E3 in Figure 11 did not show a significant downward trend after the 10-year return period (p.451-p.452).

Response: We have revised as follow (L455-457):

Considering the lags in peak time, the peak reduction rates of E3-E6 all exhibit decreasing trends for return periods greater than 10 years, the volume reduction rate was not significant.